# A cross-sectional study of physicians on fluoride-related beliefs and practices, and experiences with fluoride-hesitant caregivers

**Tiffany Bass[1], Courtney M. Hill[2], Jennifer L. Cully[3,4], Sophie R. Li[2], Donald L. Chi[1,2]***

**1** Department of Health Systems and Population Health, University of Washington, Seattle, Washington, United States of America, **2** Department of Oral Health Sciences, University of Washington, Seattle, Washington, United States of America, **3** Division of Oral Health, Children's National Hospital, Washington, DC, United States of America, **4** Department of Pediatrics, George Washington University, Washington, DC, United States of America

* dchi@uw.edu

**Data Availability Statement:** All relevant data are within the manuscript and its Supporting Information files.

## Abstract

The goal of this study was to describe medical providers' fluoride-related beliefs and practices, experiences with fluoride-hesitant caregivers, and barriers to incorporating oral health activities into their practice. In this cross-sectional study, we specifically tested the hypothesis of whether these factors differed between pediatric and family medicine providers. A 39-item online survey was administered to a convenience sample of pediatric and family medicine providers in Washington state and Ohio (U.S.A.). Responses to the fluoride survey were compared between pediatric and family medicine providers with a chi-square test (α = 0.05). Of the 354 study participants, 45% were pediatric providers and 55% were family medicine providers. About 61.9% of providers believed fluoridated water was highly effective at preventing tooth decay while only 29.1% believed prescription fluoride supplements were highly effective. Nearly all providers recommend over-the-counter fluoride toothpaste (87.3%), 44.1% apply topical fluoride in clinic, and 30.8% prescribe fluoride supplements. Most providers reported fluoride hesitancy was a small problem or not a problem (82.5%) and the most common concerns patients raise about fluoride were similar to those raised about vaccines. Lack of time was the most commonly reported barrier to incorporating oral health into practice, which was more commonly reported by family medicine providers than pediatric providers (65.6% vs. 50.3%; p = .005). Pediatric and family medicine providers have early and frequent access to children before children visit a dentist. Improving the use of fluorides through children's medical visits could improve pediatric oral health and reduce oral health inequities, especially for vulnerable populations at increased risk for tooth decay.

## Introduction

Tooth decay (dental caries or cavities) is largely preventable but remains the most common chronic disease in children and disproportionately impacts those from low-income families [1]. Downstream consequences of childhood tooth decay include pain, hospitalizations, school

**Funding:** This work was supported under the National Institute of Dental and Craniofacial Research (NIH/NIDCR) grant no. R01DE026741 (PI: DLC) and by Llyod and Kay Chapman Endowed Chair for Oral Health. The funding sources had no role in the study design, collection, analysis or interpretation of data; in the writing of the report; or in the decision to submit the article for publication.

**Competing interests:** The authors have declared that no competing interests exist.

absences, poor academic performance, and lower quality of life [2–4]. In line with anticipatory guidance recommended by the American Academy of Pediatric Dentistry to prevent tooth decay, the American Academy of Pediatrics and the American Academy of Family Physicians recommend that children receive their first dental visit by no later than age 12 months [5–7]. However, less than 2% of low-income children meet this recommendation [8]. In contrast, most children will see a medical provider for as many as seven well-child visits before age 12 months [9]. Thus, medical providers play a critical role in helping to ensure optimal oral health for children in early life [10]. However, medical providers experience barriers to integrating oral health-related anticipatory guidance and preventive oral health treatments like topical fluoride into practice [11].

Fluoride is the cornerstone of tooth decay prevention efforts. It is safe and effective [12]. There are multiple fluoride modalities, including topical fluoride (applied by a health care provider), prescription fluoride supplements, over-the-counter fluoride toothpaste and rinses, and fluoridated water. Fluoride supplements are indicated for children at high risk for tooth decay and those who live in an area without fluoridated water [13]. The American Academy of Pediatrics suggests that pediatricians 1) recommend fluoridated toothpaste to children at the eruption of the first tooth; and 2) apply topical fluoride to high risk children under age 6 years [14]. In addition, the American Academy of Pediatrics and the American Academy of Family Physicians guidelines support water fluoridation and recommend that medical providers determine the need for fluoride supplements based on water fluoridation status of the community in which the child lives [14,15].

There are multiple documented barriers to medical providers incorporating oral health-related anticipatory guidance and fluoride application into practice. Previous work suggests oral health training during medical school and residency is insufficient in shaping future provider behaviors [16–18]. A study in Hawaii reported that a smaller proportion of family medicine providers were familiar with the American Dental Association's recommendation for daily fluoride supplementation compared to pediatric providers (53% vs. 76%) [19]. Studies among pediatricians and family medicine providers in Saudi Arabia and in Canada have found that about one-half of providers report that lack of time is a barrier to carrying out oral health activities [18,20]. Similarly, medical providers may believe that providing treatments like fluoride is outside the scope of medical practice. Among pediatricians and family medicine providers in Saudi Arabia, pediatric providers had more positive attitudes about the need for medical providers to provide pediatric dental checks compared to family medicine providers [20].

Another potential barrier is discomfort communicating with caregivers who are hesitant about fluoride, which is a behavior linked to vaccine hesitancy [21]. Vaccine hesitant parents may raise issues that many providers feel ill-equipped to respond to because of inadequate training [22]. Physicians may request that vaccine-hesitant patients leave their practice due to frustration with patient pushback [23]. Research among pediatricians in Connecticut suggests that at least 30% of providers have dismissed families because of their refusal to immunize [24] and findings from a national survey of pediatricians in 2012 and 2013 showed that 12% to 21% of pediatricians reported always or often dismissing families who refused vaccines from their practice [25,26]. Dismissal for immunization refusal is discouraged by the Centers for Disease Control and Prevention and the American Academy of Pediatrics [23].

To date, there has been no research on medical providers' experiences with topical fluoride hesitancy, but it is increasingly relevant because fluoride hesitancy is associated with vaccine hesitancy and both behaviors may be increasing in the United States [27]. Furthermore, it is unknown whether providers' experiences with fluoride hesitancy differ by medical provider type. The goals of this study were to describe fluoride beliefs and practices, experiences with fluoride-hesitant caregivers, and barriers to incorporating oral health activities into practice

among medical providers. We also determine whether each differed between pediatric providers and family medicine providers. Because medical providers play a critical role in tooth decay prevention, the findings from this study are expected to be relevant in improving fluoride policies in medical practice.

## Materials and methods

### Study design, setting, and participants

This cross-sectional study focused on four types of medical providers in Washington state and Ohio: Medical Doctors (MDs), Osteopathic Doctors (DOs), Physician Assistants (PAs), and Nurse Practitioners (NPs). In Washington, a convenience sample of participants was recruited via email from four sources: medical directors at the Community Health Plan of Washington, medical providers at Community Health Plan of Washington (n = 91), members of the Washington Academy of Family Physicians, and all MDs with an active license in Washington (n = 20,868 active MDs with an active email address). In Ohio, participants were recruited among pediatric medical providers who subscribed to the Cincinnati Children's Hospital Medical Center Physician Outreach and Engagement newsletter. Medical providers invited to the study were asked to forward our study email to other medical colleagues who might be interested. To be included in the study, there were two eligibility criteria. First, participants had to treat at least one patient under age 18 years. Second, participants had to be a pediatric- or family medicine-focused provider. Informed consent was obtained from all participants with survey initiation as agreement to participate in the research (there was no written or verbal consent obtained). The study was classified as exempt by the Institutional Review Boards at University of Washington (IRB ID: STUDY00012846) and Cincinnati Children's Hospital Medical Center (IRB ID: 2021–0228).

### Data collection

From March 1, 2021 to September 30, 2021, medical providers were contacted via email with a link to an anonymous 39-item Research Electronic Data Capture (REDCap) questionnaire [see S1 File]. The survey included five sections: provider demographic and practice characteristics, beliefs on fluoride effectiveness, current fluoride-related practices, experiences with fluoride- and vaccine-hesitant caregivers, and barriers to incorporating oral health into practice.

### Variables

**Provider type.** The main grouping variable in this study was self-reported provider type. The two groups were pediatric-focused provider vs. family medicine-focused provider.

**Demographic, professional, and practice characteristics.** The questionnaire included nine items on demographic, professional, and practice characteristics. Demographic characteristics included four items: age, gender, race, and ethnicity. Professional and practice characteristics included five items: provider type (physician, nurse practitioner, physician assistant), years since completing residency or clinical training, practice setting (university or hospital, community health center or public health clinic, private practice, military clinic), % of practice patients insured by Medicaid (0%, 1–25%, 26–50%, 51–75%, >75%), and whether they practiced in an area with fluoridated water (yes/no/unknown).

**Beliefs on fluoride effectiveness.** Providers' beliefs about fluoride effectiveness in preventing cavities were assessed for five fluoride modalities: fluoride (in general), topical fluoride, prescription fluoride supplements, over-the-counter fluoride toothpaste, and fluoridated water. Each item had four response options: very effective, effective, somewhat effective, and not effective.

**Fluoride-related practices.**   We assessed current practices for three fluoride modalities: topical fluoride application, fluoride supplement prescription, and over-the-counter fluoride toothpaste recommendation. Providers were asked whether they apply, prescribe, or recommend each fluoride modality (yes/no). Among providers who responded yes, we asked at what patient age they start applying, prescribing, or recommending the fluoride modality (<6 months, 6 months to ≤3 years, >3 years to ≤6 years, >6 years). When providers indicated they did not apply, prescribe, or recommend the fluoride modality, we provided a checklist to assess the reason(s) why. The checklist included options such as "my patients live in an area with fluoridated water", "my patients are low risk for cavities", and an open-ended option where providers could write-in other reasons. All open-ended responses were reviewed and then grouped into new categories.

**Experiences with fluoride and vaccine hesitancy.**   To measure providers' experiences with fluoride- and vaccine-hesitant caregivers, we used five items for each fluoride and vaccines (ten items total). The five items were "how often do parents/patients ask you for advice about fluoride/vaccines? (never, sometimes, often, always)", "what issues do patients/parents raise about fluoride/vaccines?", "how problematic is fluoride/vaccine hesitancy among patients in your practice? (a big problem, a medium-sized problem, a small problem, not a problem at all)", "over time, fluoride/vaccine hesitancy among patients in your practice has (gotten worse, stayed the same, gotten better, I don't know)", and "when you encounter a parent who is hesitant about fluoride/vaccines, how comfortable are you talking to the parent about changing their mind about fluoride/vaccines? (extremely comfortable, somewhat comfortable, somewhat uncomfortable, extremely uncomfortable)". The item on issues patients/parents raise was only presented to providers who indicated that parents/patients at least sometimes ask for advice and was a checklist that consisted of four issues that patients/parents raise (i.e., side effects, unsure of whether their child needs it, conflicting advice from another healthcare provider, cost).

**Barriers to incorporating oral health into practice.**   An item checklist was used to identify barriers to incorporating oral health activities into practice. Response options included "lack of time", "need to address other more important issues during medical visits", "lack of knowledge", "lack of parent/patient interest", "belief that oral health activities should be performed by dentists", "lack of dentists in the area for referral", and "lack of reimbursement". An open-ended response option was also included.

## Statistical methods

First, provider demographic, professional, and practice characteristics were summarized as mean and standard deviation (SD) or n and %. Then, fluoride survey responses were summarized for the total study population and separately by provider type (pediatric providers vs. family medicine providers). Survey responses were not further stratified by provider type categories (physician vs. NP vs. PA) because there were few responses from NPs and PAs. Differences in fluoride survey responses by provider type were tested using the t-test or chi-square test ($\alpha = 0.05$). Pairwise deletion was used to account for missing values in the analysis. All statistical analyses were conducted with R version 4.1.2.

## Results

Demographic, professional, and practice characteristics

A total of 777 individuals accessed the survey and 657 initiated the survey. After removing surveys from individuals who indicated they were neither a pediatrician nor a family medicine provider (n = 239), did not treat patients under 18 years (n = 13), and those with missing data

on provider type (n = 51), the final sample size was 354. Most providers screened out based on not being a pediatric- or family medicine-focused provider reported that they were dentists or pediatric medical specialists (e.g., emergency medicine, pulmonology).

Pediatric providers constituted 44.9% of the study sample (n = 159) and family medicine providers constituted 55.1% (n = 195) (**Table 1**). The mean age of the sample was 48.7 years (SD 19.6), most participants were women (65.0%), white (81.4%), and non-Hispanic (93.5%). Pediatric providers and family providers were similar across all other measured demographic characteristics. Almost all providers in the study were physicians (95.5%), 2.8% were NPs, and 1.7% were PAs. Pediatric providers tended to have a longer number of years since completing residency/clinical training than family medicine providers (66.7% vs. 54.3% had been practicing for more than 10 years; p = .01). Pediatric providers more commonly practiced in a private setting compared to family medicine providers but less commonly in community health centers or public health clinics compared to family medicine providers (55.3% vs. 20.0% and 20.1% vs. 49.7%; p < .001). The majority of both provider types reported that less than 50% of patients in their practice were publicly insured (e.g., by Medicaid) (61.3%) and that they practiced in an area with fluoridated water (66.9%).

## Beliefs on fluoride effectiveness

About 75.7% of providers believed that fluoride, in general, was very effective in preventing cavities (**Table 2**). Fluoridated water and topical fluoride were believed to be very effective by 61.9% and 48.9% of providers, respectively. Only 29.1% and 28.5% of providers believed that prescription fluoride supplements and OTC fluoride toothpaste were very effective. Larger proportions of pediatric providers believed fluoride was very effective, in general, and across all modalities, but only beliefs about the effectiveness of fluoride in general differed significantly by provider type (82.4% vs. 70.3%; p = .04).

## Fluoride-related practices

About 44.1% of providers reported applying topical fluoride in clinical practice, 30.8% reported prescribing fluoride supplements, and 87.3% reported recommending OTC fluoride toothpaste (**Table 3**). A significantly larger proportion of pediatric providers reported recommending OTC fluoride toothpaste than family medicine providers (92.5% vs. 83.1%; p = .01). There was no difference by provider type in the proportions who applied topical fluoride (p = .24) or prescribed fluoride supplements (p = .60).

## Experiences with fluoride and vaccine hesitancy

Only 6.2% of providers reported that parents/patients always or often ask for advice about fluoride while 76.5% reported that patients always or often ask for advice about vaccines (**Table 4**). A significantly larger proportion of family medicine providers reported that parents/patients never ask for advice about fluoride compared to pediatric providers (49.2% vs. 21.3%; p = .002). The most common issues parents/patients raised about fluoride were similar to the issues raised about vaccines: being unsure if child needs it, side effects, and conflicting advice from another healthcare provider. Almost all (82.5%) of providers reported that fluoride hesitancy was a small problem or not a problem, 7.9% thought that it had gotten worse, and 82.5% reported that they were somewhat or extremely comfortable talking to hesitant patients about fluoride. About 51.3% of providers reported that vaccine hesitancy was a medium-sized or big problem, 48.3% reported that vaccine hesitancy had gotten worse, and 96.9% reported feeling somewhat or extremely comfortable talking to patients about vaccine hesitancy.

**Table 1. Demographic, professional, and practice characteristics of pediatric (n = 159) and family medicine providers (n = 195) in study.**

| Demographic, professional, and practice characteristics | All providers (N = 354) Mean (SD) or n (%) | Pediatric provider (n = 159) Mean (SD) or n (%) | Family medicine provider (n = 195) Mean (SD) or n (%) | P-value[2] |
|---|---|---|---|---|
| **Age (years)** | 48.7 (19.6) | 47.4 (12.0) | 49.8 (24.3) | .29 |
| **Gender** | | | | .30 |
| Man | 114 (32.2%) | 45 (28.3%) | 69 (35.4%) | |
| Woman | 230 (65.0%) | 111 (69.8%) | 119 (61.0%) | |
| Non-binary | 2 (0.6%) | 1 (0.6%) | 1 (0.5%) | |
| Missing | 8 (2.3%) | 2 (1.3%) | 6 (3.1%) | |
| **Race** | | | | .07 |
| American Indian or Alaska Native | 2 (0.6%) | 0 (0.0%) | 2 (1.0%) | |
| Asian | 37 (10.5%) | 20 (12.6%) | 17 (8.7%) | |
| Black or African American | 9 (2.5%) | 6 (3.8%) | 3 (1.5%) | |
| Native Hawaiian or Pacific Islander | 1 (0.3%) | 0 (0.0%) | 1 (0.5%) | |
| White | 288 (81.4%) | 130 (81.8%) | 158 (81.0%) | |
| Multi-racial | 5 (1.4%) | 1 (0.6%) | 4 (2.1%) | |
| Other[1] | 6 (1.7%) | 0 (0.0%) | 6 (3.1%) | |
| Missing | 6 (1.7%) | 2 (1.3%) | 4 (2.1%) | |
| **Hispanic** | | | | .43 |
| Yes | 18 (5.1%) | 6 (3.8%) | 12 (6.2%) | |
| No | 331 (93.5%) | 151 (95.0%) | 180 (92.3%) | |
| **Provider type** | | | | .23 |
| Physician | 338 (95.5%) | 155 (97.5%) | 183 (93.8%) | |
| Nurse Practitioner | 10 (2.8%) | 3 (1.9%) | 7 (3.6%) | |
| Physician Assistant | 6 (1.7%) | 1 (0.6%) | 5 (2.6%) | |
| **Time since completing residency/clinical training** | | | | .01 |
| 0–5 years | 89 (25.1%) | 27 (17.0%) | 62 (31.8%) | |
| 6–10 years | 53 (15.0%) | 26 (16.4%) | 27 (13.8%) | |
| 11–20 years | 82 (23.2%) | 47 (29.6%) | 35 (17.9%) | |
| 21–30 years | 83 (23.4%) | 39 (24.5%) | 44 (22.6%) | |
| > 30 years | 47 (13.3%) | 20 (12.6%) | 27 (13.8%) | |
| **Practice setting** | | | | < .001 |
| University or hospital | 59 (16.7%) | 34 (21.4%) | 25 (12.8%) | |
| Community health center/public health clinic | 129 (36.4%) | 32 (20.1%) | 97 (49.7%) | |
| Private practice | 127 (35.9%) | 88 (55.3%) | 39 (20.0%) | |
| Other | 38 (10.7%) | 5 (3.1%) | 33 (16.9%) | |
| **Patients in practice insured by Medicaid** | | | | .21 |
| 0% | 9 (2.5%) | 3 (1.9%) | 6 (3.1%) | |
| 1–25% | 122 (34.5%) | 64 (40.3%) | 58 (29.7%) | |
| 26–50% | 86 (24.3%) | 39 (24.5%) | 47 (24.1%) | |
| 51–75% | 75 (21.2%) | 28 (17.6%) | 47 (24.1%) | |
| > 75% | 59 (16.7%) | 23 (14.5%) | 36 (18.5%) | |
| Missing | 3 (0.8%) | 2 (1.3%) | 1 (0.5%) | |
| **Practices in an area with fluoridated water** | | | | .03 |
| Yes | 237 (66.9%) | 117 (73.6%) | 120 (61.5%) | |
| No | 91 (25.7%) | 34 (21.4%) | 57 (29.2%) | |
| Unknown | 20 (5.6%) | 5 (3.1%) | 15 (7.7%) | |

*(Continued)*

**Table 1.** (Continued)

| Demographic, professional, and practice characteristics | All providers (N = 354) Mean (SD) or n (%) | Pediatric provider (n = 159) Mean (SD) or n (%) | Family medicine provider (n = 195) Mean (SD) or n (%) | P-value[2] |
|---|---|---|---|---|
| Missing | 6 (1.7%) | 3 (1.9%) | 3 (1.5%) | |

SD, standard deviation.

[1]Other race includes individuals who identified as Mexican-American, Latino, Latinx, or selected "other" race and did not specify further.

[2]P-values were generated using a t-test for age and a chi-square test for categorical variables. Missing values were removed before testing.

**Table 2.** Pediatric (n = 159) and family medicine providers (n = 195) beliefs about the effectiveness of fluoride.

| Beliefs about the effectiveness of fluoride modalities in preventing cavities | All providers (N = 354) n (%) | Pediatric provider (n = 159) n (%) | Family medicine provider (n = 195) n (%) | P-value[1] |
|---|---|---|---|---|
| **Fluoride, in general** | | | | .04 |
| Very effective | 268 (75.7%) | 131 (82.4%) | 137 (70.3%) | |
| Effective | 77 (21.8%) | 26 (16.4%) | 51 (26.2%) | |
| Somewhat effective | 5 (1.4%) | 1 (0.6%) | 4 (2.1%) | |
| Not effective | 2 (0.6%) | 0 (0.0%) | 2 (1.0%) | |
| Missing | 2 (0.6%) | 1 (0.6%) | 1 (0.5%) | |
| **Topical fluoride** | | | | .07 |
| Very effective | 173 (48.9%) | 88 (55.3%) | 85 (43.6%) | |
| Effective | 149 (42.1%) | 62 (39.0%) | 87 (44.6%) | |
| Somewhat effective | 27 (7.6%) | 7 (4.4%) | 20 (10.3%) | |
| Not effective | 2 (0.6%) | 1 (0.6%) | 1 (0.5%) | |
| Missing | 3 (0.8%) | 1 (0.6%) | 2 (1.0%) | |
| **Prescription fluoride supplements** | | | | |
| Very effective | 103 (29.1%) | 52 (32.7%) | 51 (26.2%) | .10 |
| Effective | 136 (38.4%) | 63 (39.6%) | 73 (37.4%) | |
| Somewhat effective | 74 (20.9%) | 31 (19.5%) | 43 (22.1%) | |
| Not effective | 31 (8.8%) | 8 (5.0%) | 23 (11.8%) | |
| Missing | 10 (2.8%) | 5 (3.1%) | 5 (2.6%) | |
| **OTC fluoride toothpaste** | | | | .09 |
| Very effective | 101 (28.5%) | 53 (33.3%) | 48 (24.6%) | |
| Effective | 156 (44.1%) | 72 (45.3%) | 84 (43.1%) | |
| Somewhat effective | 86 (24.3%) | 30 (18.9%) | 56 (28.7%) | |
| Not effective | 9 (2.5%) | 3 (1.9%) | 6 (3.1%) | |
| Missing | 2 (0.6%) | 1 (0.6%) | 1 (0.5%) | |
| **Fluoridated water** | | | | .05 |
| Very effective | 219 (61.9%) | 108 (67.9%) | 111 (56.9%) | |
| Effective | 104 (29.4%) | 42 (26.4%) | 62 (31.8%) | |
| Somewhat effective | 26 (7.3%) | 9 (5.7%) | 17 (8.7%) | |
| Not effective | 5 (1.4%) | 0 (0.0%) | 5 (2.6%) | |

OTC, over-the-counter.

[1]P-values were generated using a chi-square test and missing values were removed before testing.

**Table 3. Fluoride-related practices by fluoride modality among pediatric (n = 159) and family medicine providers (n = 195).**

| Current fluoride-related practices by fluoride modality | All providers (N = 354) n (%) | Pediatric provider (n = 159) n (%) | Family medicine provider (n = 195) n (%) | P-value[1] |
|---|---|---|---|---|
| **Applies topical fluoride** | | | | .24 |
| Yes | 156 (44.1%) | 64 (40.3%) | 92 (47.2%) | |
| No | 194 (54.8%) | 93 (58.5%) | 101 (51.8%) | |
| Missing | 4 (1.1%) | 2 (1.3%) | 2 (1.0%) | |
| **Recommended age to start topical fluoride application** *Among providers who indicated they apply topical fluoride* | n = 156 | n = 64 | n = 92 | |
| <6 months | 9 (5.8%) | 3 (4.7%) | 6 (6.5%) | |
| 6 months to ≤3 years | 137 (87.8%) | 59 (92.2%) | 78 (84.8%) | |
| 3 to ≤6 years | 7 (4.5%) | 1 (1.6%) | 6 (6.5%) | |
| >6 years | 2 (1.3%) | 1 (1.6%) | 1 (1.1%) | |
| Missing | 1 (0.6%) | 0 (0.0%) | 1 (1.1%) | |
| **Reasons for not applying topical fluoride** *Among providers who indicated they do not apply topical fluoride* | n = 194 | n = 93 | n = 101 | |
| Administrative barriers | 76 (39.2%) | 35 (37.6%) | 41 (40.6%) | |
| Patients live in an area with fluoridated water | 54 (27.8%) | 31 (33.3%) | 23 (22.8%) | |
| Patients receive this care from dentists | 47 (24.2%) | 17 (18.3%) | 30 (29.7%) | |
| Patients are at low risk for cavities | 14 (7.2%) | 11 (11.8%) | 3 (3.0%) | |
| There is no longer a need | 1 (0.5%) | 0 (0.0%) | 1 (1.0%) | |
| Many of my patients are hesitant | 5 (2.6%) | 2 (2.2%) | 3 (3.0%) | |
| Other (e.g., works in a subspecialty or urgent care) | 27 (13.9%) | 19 (20.4%) | 8 (7.9%) | |
| **Prescribes fluoride supplements** | | | | .60 |
| Yes | 109 (30.8%) | 46 (28.9%) | 63 (32.3%) | |
| No | 244 (68.9%) | 112 (70.4%) | 132 (67.7%) | |
| Missing | 1 (0.3%) | 1 (0.6%) | 0 (0.0%) | |
| **Recommended age to start prescribing fluoride supplements** *Among providers who indicated they prescribe fluoride supplements* | n = 109 | n = 46 | n = 63 | |
| <6 months | 8 (7.3%) | 3 (6.5%) | 5 (7.9%) | |
| 6 months to ≤3 years | 98 (89.9%) | 42 (91.3%) | 56 (88.9%) | |
| 3 to ≤6 years | 3 (2.8%) | 1 (2.2%) | 2 (3.2%) | |
| >6 years | 0 (0.0%) | 0 (0.0%) | 0 (0.0%) | |
| **Reasons for not prescribing fluoride supplements** *Among providers who indicated they do not prescribe fluoride supplements* | n = 244 | n = 112 | n = 132 | |
| Patients live in an area with fluoridated water | 164 (67.2%) | 81 (72.3%) | 83 (62.9%) | |
| Patients are hesitant | 30 (12.3%) | 8 (7.1%) | 22 (16.7%) | |
| Patients have barriers to filling prescriptions | 27 (11.1%) | 5 (4.5%) | 22 (16.7%) | |
| Lack of knowledge about prescription fluoride | 23 (9.4%) | 9 (8.0%) | 14 (10.6%) | |
| My patients are low risk for cavities | 11 (4.5%) | 6 (5.4%) | 5 (3.8%) | |
| There is no longer a need | 12 (4.9%) | 5 (4.5%) | 7 (5.3%) | |
| Other | 39 (16.0%) | 20 (17.9%) | 19 (14.4%) | |
| **Recommends OTC fluoride toothpaste** | | | | .01 |
| Yes | 309 (87.3%) | 147 (92.5%) | 162 (83.1%) | |
| No | 42 (11.9%) | 11 (6.9%) | 31 (15.9%) | |
| Missing | 3 (0.8%) | 1 (0.6%) | 2 (1.0%) | |
| **Recommended age to start using OTC fluoride toothpaste** *Among providers who indicated they recommend OTC fluoride toothpaste* | n = 309 | n = 147 | n = 162 | |
| <6 months | 18 (5.8%) | 9 (6.1%) | 9 (5.6%) | |

*(Continued)*

**Table 3.** (Continued)

| Current fluoride-related practices by fluoride modality | All providers (N = 354) n (%) | Pediatric provider (n = 159) n (%) | Family medicine provider (n = 195) n (%) | P-value[1] |
|---|---|---|---|---|
| 6 months to ≤3 years | 205 (66.3%) | 106 (72.1%) | 99 (61.1%) | |
| 3 to ≤6 years | 74 (23.9%) | 29 (19.7%) | 45 (27.8%) | |
| >6 years | 12 (3.9%) | 3 (2.0%) | 9 (5.6%) | |
| **Reasons for not recommending OTC fluoride toothpaste** *Among providers who indicated they do not recommend OTC fluoride toothpaste* | *n = 42* | *n = 11* | *n = 31* | |
| Patients live in an area with fluoridated water | 14 (33.3%) | 2 (18.2%) | 12 (38.7%) | |
| Never specifies whether toothpaste has fluoride | 10 (23.8%) | 1 (9.1%) | 9 (29.0%) | |
| Many of my patients are hesitant | 4 (9.5%) | 0 (0.0%) | 4 (12.9%) | |
| There is no longer a need | 2 (4.8%) | 0 (0.0%) | 2 (6.5%) | |
| My patients are low risk for cavities | 0 (0.0%) | 0 (0.0%) | 0 (0.0%) | |
| Other | 14 (33.3%) | 8 (72.7%) | 6 (19.3%) | |

OTC, over-the-counter.

[1]P-values were generated using a chi-square test and missing values were removed before testing.

### Barriers to incorporating oral health into practice

The most common reported barriers to incorporating oral health into practice were lack of time (58.8% of all providers), the need to address more important issues during medical visits (50.0%), lack of knowledge (33.1%), and lack of parent/patient interest (27.4%) (**Table 5**). Family medicine providers were significantly more likely to report lack of time as a barrier compared to pediatric providers (65.6% vs. 50.3%; p = .005). There were no other significant differences in reported barriers by provider type.

## Discussion

In this study, we examined fluoride-related beliefs and practices of pediatric and family medicine providers, their clinical experiences with fluoride-hesitant caregivers, and barriers to incorporating oral health activities into practice. There were three main findings. First, most providers believed fluoride prevents cavities and while nearly all recommended fluoride toothpaste, less than one-half applied topical fluoride in clinic. Second, relatively few providers reported fluoride hesitancy being a problem in clinic though caregiver concerns about fluoride mirrored concerns raised about vaccines. Third, the most common barriers to incorporating oral health into practice were lack of time and the need to address other issues during visits.

Most providers believed in the effectiveness of fluoride. More than 75% of providers in our study believed fluoride was very effective at preventing cavities, which aligns with previous research [28]. A significantly larger proportion of pediatric providers believed fluoride was very effective compared to family medicine providers (82.4% vs. 70.3%; P = 0.04), which may be influenced by greater knowledge about fluoride. Previous research found that pediatricians were more likely to correctly answer fluoride-related questions and report greater confidence in their knowledge about fluoride than family medicine providers [29], suggesting that fluoride guidance has been disseminated differentially across medical specialties. Another explanation is that interprofessional education on oral health has focused primarily on pediatrics [30], which highlights the potential for future efforts within non-pediatrics-focused primary care specialties, including family medicine, internal medicine, and obstetrics and gynecology.

**Table 4.** Experiences with fluoride- and vaccine-hesitant parents among pediatric (n = 159) and family medicine providers (n = 195).

| Experiences with fluoride and vaccine hesitancy | All providers (N = 354) n (%) | Pediatric provider (n = 159) n (%) | Family medicine provider (n = 195) n (%) | P-value[1] |
|---|---|---|---|---|
| **Parents/patients ask for advice about fluoride**... | | | | .002 |
| Always | 0 (0.0%) | 0 (0.0%) | 0 (0.0%) | |
| Often | 22 (6.2%) | 9 (5.7%) | 13 (6.7%) | |
| Sometimes | 185 (52.3%) | 99 (62.3%) | 86 (44.1%) | |
| Never | 146 (41.2%) | 50 (31.4%) | 96 (49.2%) | |
| Missing | 1 (0.3%) | 1 (0.6%) | 0 (0.0%) | |
| **Parents/patients raise these issues about fluoride**... *Among providers who indicated parents/patients ask for advice always, often, or sometimes* | n = 207 | n = 108 | n = 99 | |
| Unsure of whether child needs it | 146 (70.5%) | 77 (71.3%) | 69 (69.7%) | |
| Side effects | 77 (37.2%) | 42 (38.9%) | 36 (36.4%) | |
| Conflicting advice from another healthcare provider | 51 (24.6%) | 33 (30.5%) | 19 (19.2%) | |
| Cost | 8 (3.9%) | 2 (0.9%) | 6 (6.1%) | |
| Other | 12 (5.8%) | 6 (0.9%) | 6 (6.1%) | |
| **Fluoride hesitancy is a**... | | | | .53 |
| Big problem | 10 (2.8%) | 3 (1.9%) | 7 (3.6%) | |
| Medium- sized problem | 47 (13.3%) | 18 (11.3%) | 29 (14.9%) | |
| Small problem | 162 (45.8%) | 77 (48.4%) | 85 (43.6%) | |
| Not a problem | 130 (36.7%) | 59 (37.1%) | 71 (36.4%) | |
| Missing | 5 (1.4%) | 2 (1.3%) | 3 (1.5%) | |
| **Over time, fluoride hesitancy has**... | | | | .54 |
| Gotten worse (more parents are refusing) | 28 (7.9%) | 12 (7.5%) | 16 (8.2%) | |
| Stayed about the same | 135 (38.1%) | 61 (38.4%) | 74 (37.9%) | |
| Gotten better (fewer parents are refusing) | 38 (10.7%) | 21 (13.2%) | 17 (8.7%) | |
| I don't know | 152 (42.9%) | 64 (40.3%) | 88 (45.1%) | |
| Missing | 1 (0.3%) | 1 (0.6%) | 0 (0.0%) | |
| **Provider comfort with fluoride hesitancy** | | | | .06 |
| Extremely comfortable | 110 (31.1%) | 56 (35.2%) | 54 (27.7%) | |
| Somewhat comfortable | 182 (51.4%) | 80 (50.3%) | 102 (52.3%) | |
| Somewhat uncomfortable | 50 (14.1%) | 22 (13.8%) | 28 (14.4%) | |
| Extremely uncomfortable | 7 (2.0%) | 0 (0.0%) | 7 (3.6%) | |
| Missing | 5 (1.4%) | 1 (0.6%) | 4 (2.1%) | |
| **Parents/ patients ask for advice about vaccines**... | | | | .03 |
| Always | 74 (20.9%) | 44 (27.7%) | 30 (15.4%) | |
| Often | 197 (55.6%) | 84 (52.8%) | 113 (57.9%) | |
| Sometimes | 78 (22.0%) | 30 (18.9%) | 48 (24.6%) | |
| Never | 1 (0.3%) | 0 (0.0%) | 1 (0.5%) | |
| Missing | 4 (1.1%) | 1 (0.6%) | 3 (1.5%) | |
| **Parents/patients raise these issues about vaccines**... *Among providers who indicated parents/patients ask for advice always, often, or sometimes* | n = 349 | n = 158 | n = 191 | |
| Side effects | 326 (93.4%) | 153 (96.8%) | 173 (90.6%) | |
| Unsure of whether child needs It | 246 (70.5%) | 114 (72.2%) | 132 (69.1%) | |
| Conflicting advice from another healthcare provider | 98 (28.1%) | 41 (25.9%) | 57 (29.8%) | |
| Cost | 17 (4.9%) | 4 (2.5%) | 13 (6.8%) | |
| Other | 39 (11.2%) | 15 (9.5%) | 24 (12.6%) | |
| **Vaccine hesitancy is a**... | | | | .52 |

*(Continued)*

**Table 4.** (Continued)

| Experiences with fluoride and vaccine hesitancy | All providers (N = 354) n (%) | Pediatric provider (n = 159) n (%) | Family medicine provider (n = 195) n (%) | P-value[1] |
|---|---|---|---|---|
| Big problem | 67 (18.9%) | 31 (19.5%) | 36 (18.5%) | |
| Medium- sized problem | 150 (42.4%) | 61 (38.4%) | 89 (45.6%) | |
| Small problem | 130 (36.7%) | 63 (39.6%) | 67 (34.4%) | |
| Not a problem | 5 (1.4%) | 3 (1.9%) | 2 (1.0%) | |
| Missing | 2 (0.6%) | 1 (0.6%) | 1 (0.5%) | |
| **Vaccine hesitancy has…** | | | | .09 |
| Gotten worse (more parents are refusing) | 171 (48.3%) | 73 (45.9%) | 98 (50.3%) | |
| Stayed about the same | 110 (31.1%) | 46 (28.9%) | 64 (32.8%) | |
| Gotten better (fewer parents are refusing) | 43 (12.1%) | 27 (17.0%) | 16 (8.2%) | |
| I don't know | 28 (7.9%) | 12 (7.5%) | 16 (8.2%) | |
| Missing | 2 (0.6%) | 1 (0.6%) | 1 (0.5%) | |
| **Provider comfort with vaccine hesitancy** | | | | .66 |
| Extremely comfortable | 252 (71.2%) | 117 (73.6%) | 135 (69.2%) | |
| Somewhat comfortable | 91 (25.7%) | 37 (23.3%) | 54 (27.7%) | |
| Somewhat uncomfortable | 4 (1.1%) | 1 (0.6%) | 3 (1.5%) | |
| Extremely uncomfortable | 5 (1.4%) | 2 (1.3%) | 3 (1.5%) | |
| Missing | 2 (0.6%) | 2 (1.3%) | 0 (0.0%) | |

[1]P-values were generated using a chi-square test and missing values were removed before testing.

Beliefs about the effectiveness of fluoride in preventing cavities differed by fluoride modality. About 61.9% of providers thought fluoridated water was very effective, whereas 48.9%, 29.1%, and 28.5% thought topical fluoride, prescription fluoride supplements, and OTC fluoride toothpaste were very effective, respectively. There are currently no studies on modality-focused fluoride beliefs among medical providers. Future studies should explore strategies to increase clinical support for different fluoride modalities, which could translate into greater knowledge among caregivers, greater alignment of preventive care messaging between medical and dental health care providers, and improvements in children's oral health [31].

**Table 5. Barriers to incorporating oral health into practice among pediatric (n = 159) and family medicine providers (n = 195).**

| Barriers to incorporating oral health activities | All providers (N = 354) n (%)[1] | Pediatric provider (n = 159) n (%) | Family medicine provider (n = 195) n (%) | P-value[2] |
|---|---|---|---|---|
| Lack of time | 208 (58.8%) | 80 (50.3%) | 128 (65.6%) | .005 |
| Need to address other more important issues during medical visits | 177 (50.0%) | 71 (44.7%) | 106 (54.4%) | .09 |
| Lack of knowledge | 117 (33.1%) | 51 (32.1%) | 66 (33.8%) | .81 |
| Lack of parent/patient interest | 97 (27.4%) | 39 (24.5%) | 58 (29.7%) | .33 |
| Lack of reimbursement | 45 (12.7%) | 26 (16.4%) | 19 (9.7%) | .09 |
| Belief these activities should be performed by dentists | 58 (16.4%) | 24 (15.1%) | 34 (17.4%) | .65 |
| Lack of dentists in the area for referral | 29 (8.2%) | 13 (8.2%) | 16 (8.2%) | .99 |
| Other | 30 (8.5%) | 15 (9.4%) | 15 (7.7%) | .69 |

[1] The columns do not sum to 100% because participants were instructed to select each barrier that applies.

[2] P-values were generated using a chi-square test.

Providers reported incorporating fluoride into their current practices and recommending different types of fluoride to prevent cavities. About 87.3% of all providers recommended OTC fluoride toothpaste, 44.1% reported applying topical fluoride in clinic, and 30.8% prescribed fluoride supplements. A greater proportion of pediatric providers in our study reported recommending OTC fluoride toothpaste than family medicine providers (92.5% vs. 83.1%). This difference may be influenced by different training and academic practice guidelines between pediatric and family medicine providers. The American Academy of Pediatrics recommends that fluoride toothpaste be used at the eruption of the first tooth [6,14]. In a study of Canadian pediatricians and family physicians, pediatricians received more training and attended more Continuing Medical Education courses on oral health than family physicians [18]. In general, previous research has suggested that oral health training during medical school and residency is insufficient [16,17]. Future research should explore ways to improve fluoride-related practices in medical settings.

Providers reported their experiences with caregivers asking for advice about fluoride and vaccines, as well as their comfortability addressing fluoride hesitancy and vaccine hesitancy. A greater proportion of pediatric providers reported parents asking about fluoride (68%) compared to family medicine providers (50.8%). We observed a similar trend with vaccines. However, a greater proportion of providers reported that patients ask about vaccines compared to fluoride (98.5% vs. 58.5%). This difference may be because vaccines are a routine part of primary medical care as well as an emphasis on vaccines that increased during the COVID-19 pandemic [32], leading to greater public awareness and sensitivities about vaccines. In addition, about 96.9% of providers reported feeling somewhat or extremely comfortable talking to patients about vaccine hesitancy, while only 82.5% reported feeling somewhat or extremely comfortable talking to patients about fluoride hesitancy. No known studies have focused on medical providers' comfort addressing fluoride hesitancy, though a study of general dentists and pediatric dentists in Washington state found fluoride refusal to be a significant problem in clinical practice, with large numbers of dentists feeling uncomfortable talking to caregivers who refused fluoride [27]. Given the associations between vaccine and fluoride hesitancy [21], strategies for addressing vaccine hesitancy may be applicable in addressing fluoride hesitancy. Such strategies include seeking to understand patient concerns and providing clarification in a nonjudgmental way [27].

The most common reported barrier to incorporating oral health into practice among providers was lack of time. About 58.8% of all providers reported lack of time as a barrier, which aligns with previous research [17,18,20]. In our study, family medicine providers were significantly more likely to report lack of time as a barrier compared to pediatric providers (65.6% vs. 50.3%). This barrier may be related to scope of practice. While pediatric providers focus on children, family medicine providers treat acute and chronic conditions for patients of all ages. Practice breadth may factor into the lack of time reported by family medicine providers, thus limiting their ability to incorporate oral health into visits. However, prior findings have indicated that medical providers express a positive attitude and willingness to engage in promoting the oral health of children [16,17]. Several programs have documented successful implementation of oral health activities in medical settings. One study found that including fluoride application in patient visits did not affect clinic flow and that adherence was high once a protocol was developed [33]. Another study demonstrated that brief education interventions and stop measures in electronic health record systems improved fluoride applications [34]. To overcome administrative barriers, some have found that dental hygienists incorporated into medical settings helped improve consistency and address lack of oral health knowledge among medical providers [35]. Future research should explore feasible ways oral health messaging can be meaningfully incorporated into practice.

There are two study limitations. First, the use of convenience sampling in two states limits generalizability. Second, our sample consisted mostly of white physicians. There were few responses from nurse practitioners and physician assistants as well as non-white providers. Future studies should focus on recruiting a more diverse sample of providers, which is especially important given the propensity of minority providers to treat minority children who are disproportionately affected by tooth decay.

## Conclusion

Fluoride is important for tooth decay prevention in children. Medical providers play an important role in disseminating fluoride-related knowledge in clinical settings to caregivers, especially those who are hesitant about fluoride. Future efforts should focus on improving medical provider awareness about the benefits of various fluoride modalities and the growing prevalence of fluoride hesitancy among caregivers as well as addressing barriers to incorporating oral health activities in clinic.

## Supporting information

**S1 Data. Limited use data provider fluoride survey 20240709.**
(CSV)

**S1 File. Provider survey on fluoride REDCap.**
(PDF)

## Acknowledgments

We would like to thank all the participating providers for their time in completing the survey.

## Author Contributions

**Conceptualization:** Tiffany Bass, Donald L. Chi.

**Data curation:** Tiffany Bass.

**Formal analysis:** Tiffany Bass, Courtney M. Hill, Jennifer L. Cully.

**Methodology:** Tiffany Bass, Donald L. Chi.

**Writing – original draft:** Tiffany Bass, Courtney M. Hill, Jennifer L. Cully, Sophie R. Li, Donald L. Chi.

**Writing – review & editing:** Tiffany Bass, Jennifer L. Cully, Donald L. Chi.

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
