## [Decision Letter · Decision Letter 0]

16 Jun 2024

PONE-D-24-16749A cross-sectional study of pediatric and family medicine providers on fluoride-related beliefs and practices, and experiences with fluoride-hesitant caregiversPLOS ONE

Dear Dr. Chi,

Thank you for submitting your manuscript to PLOS ONE. After careful consideration, we feel that it has merit but does not fully meet PLOS ONE’s publication criteria as it currently stands. Therefore, we invite you to submit a revised version of the manuscript that addresses the points raised during the review process.

Please include the following items when submitting your revised manuscript:</br30june,>A rebuttal letter that responds to each point raised by the academic editor and reviewer(s). You should upload this letter as a separate file labeled 'Response to Reviewers'.A marked-up copy of your manuscript that highlights changes made to the original version. You should upload this as a separate file labeled 'Revised Manuscript with Track Changes'.An unmarked version of your revised paper without tracked changes. You should upload this as a separate file labeled 'Manuscript'.If applicable, we recommend that you deposit your laboratory protocols in protocols.io to enhance the reproducibility of your results. Protocols.io assigns your protocol its own identifier (DOI) so that it can be cited independently in the future. For instructions see: https://journals.plos.org/plosone/s/submission-guidelines#loc-laboratory-protocols. Additionally, PLOS ONE offers an option for publishing peer-reviewed Lab Protocol articles, which describe protocols hosted on protocols.io. Read more information on sharing protocols at https://plos.org/protocols?utm_medium=editorial-email&utm_source=authorletters&utm_campaign=protocols.

We look forward to receiving your revised manuscript.

Kind regards,

Hadi Ghasemi

Academic Editor

PLOS ONE

Journal Requirements:

Reviewers' comments:

Reviewer's Responses to Questions

**Comments to the Author**

1. Is the manuscript technically sound, and do the data support the conclusions?

Reviewer #1: Yes

Reviewer #2: Yes

Reviewer #3: Yes

2. Has the statistical analysis been performed appropriately and rigorously? 

Reviewer #1: Yes

Reviewer #2: Yes

Reviewer #3: Yes

3. Have the authors made all data underlying the findings in their manuscript fully available?

Reviewer #1: Yes

Reviewer #2: Yes

Reviewer #3: Yes

4. Is the manuscript presented in an intelligible fashion and written in standard English?

Reviewer #1: Yes

Reviewer #2: Yes

Reviewer #3: Yes

5. Review Comments to the Author

Reviewer #1: As a reviewer, I find several strengths and areas for improvement in the study on fluoride-related beliefs and practices among pediatric and family medicine providers.

Strengths:

Clear Objectives and Findings Presentation: The study clearly outlines its objectives and presents its findings in a structured manner, making it easy for readers to understand the research aims and outcomes.

Relevant and Timely Topic: With increasing concerns about fluoride hesitancy among caregivers, especially mirroring vaccine hesitancy, the study addresses a pertinent issue in pediatric and family medicine practice.

Insightful Comparison between Specialties: The comparison between pediatric and family medicine providers provides valuable insights into potential differences in knowledge, beliefs, and practices regarding fluoride recommendations.

Identification of Barriers: The study effectively identifies common barriers to incorporating oral health activities into medical practice, particularly the lack of time, which is a practical concern for many healthcare providers.

Practical Implications: By highlighting the need to improve medical provider awareness about the benefits of fluoride and addressing fluoride hesitancy among caregivers, the study offers practical implications for enhancing preventive dental care in clinical settings.

Areas for Improvement:

Sampling Limitations: The use of convenience sampling in two states and the predominantly white physician sample limit the generalizability of the findings. Future studies should strive for more diverse and representative samples to ensure broader applicability of the results, especially considering the disproportionate impact of tooth decay on minority children.

Quantitative Data Analysis: While the study provides valuable quantitative data on providers' beliefs and practices, it would benefit from more robust statistical analyses to strengthen the validity and reliability of the findings.

Qualitative Insights: Incorporating qualitative methods, such as interviews or focus groups, could provide deeper insights into the reasons behind providers' beliefs, practices, and perceived barriers, enriching the understanding of fluoride-related attitudes in clinical practice.

Recommendation Strategies: While the study mentions the need to explore strategies for addressing fluoride hesitancy, it could elaborate further on specific intervention approaches or educational programs that could be implemented in medical settings to improve fluoride-related practices.

Addressing these areas for improvement could enhance the rigor and comprehensiveness of the study, thereby strengthening its contribution to the literature on preventive dental care in pediatric and family medicine.

Best Regards

Reviewer #2: Dear Author

Good time

It was a pleasure to read the article. It is generally well written and of clinical and public value. However, there are few comments that will enrich the article and increases its academic value.

Generally, it is of clinical value at the public level and tackles an important subject. It needs to be presented in a better easy way and use simpler language

Title: It is too long and not easy to understand, make it simpler and to the point

Abstract: Generally well written and presented with no serious comments.

Introduction: it is fine but too long and contains many sentences that need to be in the discussion.

1- lines 55-58 not easy to understand, needs rewriting

2-lines 73-77 do not have references!!! are they your results??

3- lines 81-85 as above

Methodology: overall is acceptable but attention is to be paid for the below points ;

1-instead of the heading : study participants it is better to be: study design, setting and participants

2- Need to make a flow chart for your methodology

3-the grouping is not clear, make them consistent throughout the study, are they two groups or more ?

4- PLEASE USE SIMPLE CLEAR LANGUAGE THAT DELIVERS THE MEANING

Results: Not easy to understand so PLEASE MAKE IT EASY

1- Lines 231-233 not clear

2- Table 1: regarding age I am not sure if the mean of the age has percentage !!!

3- Table 1: Why Hispanic is not part of the race and is a separate entity???

Discussion : acceptable with logical flow of the section but it seems to be slightly long

Conclusion: reasonable and not extravagant

Reviewer #3: Dear authors,

Thanks for sharing your work with us, some points should be kept in consideration:

1.The manuscript within the scope of the journal.

2.Both the quality and data presentation of this manuscript are good and of great importance to dentists, physicians and even patients.

3.The manuscript expands our knowledge about fluoride-related beliefs and practices

4.The title should be revised and reduced its characters ( precise & and informative)

5.The abstract should reflect the content of the article and must be with range of 250-300 words.

6.more paragraphs should be incorporated to introduction and discussion about the justification of your findings and comparison with other recent relevant studies.

7.Up to date references should be kept in your reference list and the old should be omitted.

Good Luck

6. PLOS authors have the option to publish the peer review history of their article (what does this mean?). If published, this will include your full peer review and any attached files.

Reviewer #1: **Yes: **Bassam Alsheekhly

Reviewer #2: No

Reviewer #3: **Yes: **Tahrir Aldelaimi

---

## [Author Response · Author response to Decision Letter 0]

17 Jun 2024

Dear Dr. Ghasemi,

We are pleased to submit our revised paper. Below we provide detailed responses to each of the reviewer’s comments. Responses are indicated below. Thank you very much.

Sincerely,

Donald Chi

I thank the editor for providing this opportunity to participate in the manuscript review

As a reviewer, I find several strengths and areas for improvement in the study on fluoride-related beliefs and practices among pediatric and family medicine providers.

 Thank you very much.

Strengths:

Clear Objectives and Findings Presentation: The study clearly outlines its objectives and presents its findings in a structured manner, making it easy for readers to understand the research aims and outcomes.

 Thank you.

Relevant and Timely Topic: With increasing concerns about fluoride hesitancy among caregivers, especially mirroring vaccine hesitancy, the study addresses a pertinent issue in pediatric and family medicine practice.

 We agree.

Insightful Comparison between Specialties: The comparison between pediatric and family medicine providers provides valuable insights into potential differences in knowledge, beliefs, and practices regarding fluoride recommendations.

 Thank you.

Identification of Barriers: The study effectively identifies common barriers to incorporating oral health activities into medical practice, particularly the lack of time, which is a practical concern for many healthcare providers.

 Thank you.

Practical Implications: By highlighting the need to improve medical provider awareness about the benefits of fluoride and addressing fluoride hesitancy among caregivers, the study offers practical implications for enhancing preventive dental care in clinical settings.

 Thank you.

Areas for Improvement:

Sampling Limitations: The use of convenience sampling in two states and the predominantly white physician sample limit the generalizability of the findings. Future studies should strive for more diverse and representative samples to ensure broader applicability of the results, especially considering the disproportionate impact of tooth decay on minority children.

 We agree. These are noted as the two main limitations in the Discussion section.

Quantitative Data Analysis: While the study provides valuable quantitative data on providers' beliefs and practices, it would benefit from more robust statistical analyses to strengthen the validity and reliability of the findings.

This was a descriptive study to evaluate physicians’ views on fluoride hesitancy. Since it is the first such study, our goal was to describe their views and see if there are potential differences by medical provide type. The statistical methods employed were sufficient for these goals. We look forward to using more complex statistical methods in the future to answer more complex study hypotheses.

Qualitative Insights: Incorporating qualitative methods, such as interviews or focus groups, could provide deeper insights into the reasons behind providers' beliefs, practices, and perceived barriers, enriching the understanding of fluoride-related attitudes in clinical practice.

 We agree. We hope to incorporate qualitative methods into future investigations.

Recommendation Strategies: While the study mentions the need to explore strategies for addressing fluoride hesitancy, it could elaborate further on specific intervention approaches or educational programs that could be implemented in medical settings to improve fluoride-related practices.

We do not currently have evidence-based strategies to address fluoride hesitancy in dental settings, which precludes our team from recommending strategies for use in medical settings. We do discuss the possibility of incorporating dental hygienists in medical practice settings, which has been shown to be effective in some clinics. Future work will hopefully continue to help clarify what these approaches are. 

Addressing these areas for improvement could enhance the rigor and comprehensiveness of the study, thereby strengthening its contribution to the literature on preventive dental care in pediatric and family medicine.

 Thank you again.

Best Regards 

Reviewers' comments:

Reviewer's Responses to Questions

Comments to the Author

1. Is the manuscript technically sound, and do the data support the conclusions?

Reviewer #1: Yes

Reviewer #2: Yes

Reviewer #3: Yes

2. Has the statistical analysis been performed appropriately and rigorously? 

Reviewer #1: Yes

Reviewer #2: Yes

Reviewer #3: Yes

3. Have the authors made all data underlying the findings in their manuscript fully available?

Reviewer #1: Yes

Reviewer #2: Yes

Reviewer #3: Yes

4. Is the manuscript presented in an intelligible fashion and written in standard English?

Reviewer #1: Yes

Reviewer #2: Yes

Reviewer #3: Yes

5. Review Comments to the Author

Reviewer #1: As a reviewer, I find several strengths and areas for improvement in the study on fluoride-related beliefs and practices among pediatric and family medicine providers.

Strengths:

Clear Objectives and Findings Presentation: The study clearly outlines its objectives and presents its findings in a structured manner, making it easy for readers to understand the research aims and outcomes.

Thank you.

Relevant and Timely Topic: With increasing concerns about fluoride hesitancy among caregivers, especially mirroring vaccine hesitancy, the study addresses a pertinent issue in pediatric and family medicine practice.

We agree.

Insightful Comparison between Specialties: The comparison between pediatric and family medicine providers provides valuable insights into potential differences in knowledge, beliefs, and practices regarding fluoride recommendations.

Thank you.

Identification of Barriers: The study effectively identifies common barriers to incorporating oral health activities into medical practice, particularly the lack of time, which is a practical concern for many healthcare providers.

Agreed.

Practical Implications: By highlighting the need to improve medical provider awareness about the benefits of fluoride and addressing fluoride hesitancy among caregivers, the study offers practical implications for enhancing preventive dental care in clinical settings.

Thank you.

Areas for Improvement:

Sampling Limitations: The use of convenience sampling in two states and the predominantly white physician sample limit the generalizability of the findings. Future studies should strive for more diverse and representative samples to ensure broader applicability of the results, especially considering the disproportionate impact of tooth decay on minority children.

We agree. These are noted as the two main limitations in the Discussion section.

Quantitative Data Analysis: While the study provides valuable quantitative data on providers' beliefs and practices, it would benefit from more robust statistical analyses to strengthen the validity and reliability of the findings.

This was a descriptive study to evaluate physicians’ views on fluoride hesitancy. Since it is the first such study, our goal was to describe their views and see if there are potential differences by medical provide type. The statistical methods employed were sufficient for these goals. We look forward to using more complex statistical methods in the future to answer more complex study hypotheses.

Qualitative Insights: Incorporating qualitative methods, such as interviews or focus groups, could provide deeper insights into the reasons behind providers' beliefs, practices, and perceived barriers, enriching the understanding of fluoride-related attitudes in clinical practice.

We agree. We hope to incorporate qualitative methods into future investigations.

Recommendation Strategies: While the study mentions the need to explore strategies for addressing fluoride hesitancy, it could elaborate further on specific intervention approaches or educational programs that could be implemented in medical settings to improve fluoride-related practices.

We do not currently have evidence-based strategies to address fluoride hesitancy in dental settings, which precludes our team from recommending strategies for use in medical settings. We do discuss the possibility of incorporating dental hygienists in medical practice settings, which has been shown to be effective in some clinics. Future work will hopefully continue to help clarify what these approaches are. 

Addressing these areas for improvement could enhance the rigor and comprehensiveness of the study, thereby strengthening its contribution to the literature on preventive dental care in pediatric and family medicine.

Thank you again.

Best Regards

Reviewer #2: Dear Author

Good time

It was a pleasure to read the article. It is generally well written and of clinical and public value. However, there are few comments that will enrich the article and increases its academic value.

Generally, it is of clinical value at the public level and tackles an important subject. It needs to be presented in a better easy way and use simpler language

Title: It is too long and not easy to understand, make it simpler and to the point

We shortened the title as suggested.

Abstract: Generally well written and presented with no serious comments.

Introduction: it is fine but too long and contains many sentences that need to be in the discussion.

1- lines 55-58 not easy to understand, needs rewriting

The sentence length has been reduced.

2-lines 73-77 do not have references!!! are they your results??

The sentences without references have been removed.

3- lines 81-85 as above

Ditto as above.

Methodology: overall is acceptable but attention is to be paid for the below points ;

1-instead of the heading : study participants it is better to be: study design, setting and participants

The section has been relabeled.

2- Need to make a flow chart for your methodology

We believe the methodology is straightforward and therefore chose not to include a flow chart.

3-the grouping is not clear, make them consistent throughout the study, are they two groups or more ?

There are 2 groups. This is clarified in the revised paper.

4- PLEASE USE SIMPLE CLEAR LANGUAGE THAT DELIVERS THE MEANING

Results: Not easy to understand so PLEASE MAKE IT EASY

1- Lines 231-233 not clear

These are findings directly presented in the table and are self-explanatory. Clarified that Medicaid means publicly-insured.

2- Table 1: regarding age I am not sure if the mean of the age has percentage !!!

These are mean (SD) as indicated at the top of the Table.

3- Table 1: Why Hispanic is not part of the race and is a separate entity???

Hispanic ethnicity is a separate variable from race in the U.S.

Discussion : acceptable with logical flow of the section but it seems to be slightly long

Conclusion: reasonable and not extravagant

Reviewer #3: Dear authors,

Thanks for sharing your work with us, some points should be kept in consideration:

1.The manuscript within the scope of the journal.

Thank you.

2.Both the quality and data presentation of this manuscript are good and of great importance to dentists, physicians and even patients.

Thank you very much.

3.The manuscript expands our knowledge about fluoride-related beliefs and practices

We agree.

4.The title should be revised and reduced its characters ( precise & and informative)

The title has been shortened.

5.The abstract should reflect the content of the article and must be with range of 250-300 words.

The abstract length was increased to 253 words.

6.more paragraphs should be incorporated to introduction and discussion about the justification of your findings and comparison with other recent relevant studies.

All of the available studies have been cited in both sections, and comparisons are made whenever possible.

7.Up to date references should be kept in your reference list and the old should be omitted.

The cited references are the ones that are most up to date and more importantly only relevant references are cited.

Good Luck

 Thank you!

6. PLOS authors have the option to publish the peer review history of their article (what does this mean?). If published, this will include your full peer review and any attached files.

Do you want your identity to be public for this peer review? For information about this choice, including consent withdrawal, please see our Privacy Policy.

Reviewer #1: Yes: Bassam Alsheekhly

Reviewer #2: No

Reviewer #3: Yes: Tahrir Aldelaimi

---

## [Decision Letter · Decision Letter 1]

1 Jul 2024

A cross-sectional study of physicians on fluoride-related beliefs and practices, and experiences with fluoride-hesitant caregivers

PONE-D-24-16749R1

Dear Dr. Donald L. Chi,

We’re pleased to inform you that your manuscript has been judged scientifically suitable for publication and will be formally accepted for publication once it meets all outstanding technical requirements.

Kind regards,

Hadi Ghasemi

Academic Editor

PLOS ONE

Additional Editor Comments (optional):

Reviewers' comments:

Reviewer's Responses to Questions

**Comments to the Author**

1. If the authors have adequately addressed your comments raised in a previous round of review and you feel that this manuscript is now acceptable for publication, you may indicate that here to bypass the “Comments to the Author” section, enter your conflict of interest statement in the “Confidential to Editor” section, and submit your "Accept" recommendation.

Reviewer #2: All comments have been addressed

Reviewer #3: All comments have been addressed

2. Is the manuscript technically sound, and do the data support the conclusions?

Reviewer #2: Yes

Reviewer #3: Yes

3. Has the statistical analysis been performed appropriately and rigorously? 

Reviewer #2: Yes

Reviewer #3: Yes

4. Have the authors made all data underlying the findings in their manuscript fully available?

Reviewer #2: Yes

Reviewer #3: Yes

5. Is the manuscript presented in an intelligible fashion and written in standard English?

Reviewer #2: Yes

Reviewer #3: Yes

6. Review Comments to the Author

Reviewer #2: Dear Author

You addressed my concern. The work is of significant scientific merit

Wishing you the best in your academia and clinical work

Reviewer #3: Dear authors

Thanks for sharing your work with us, the authors have performing all the suggested comments and revisions.

Good Luck

7. PLOS authors have the option to publish the peer review history of their article (what does this mean?). If published, this will include your full peer review and any attached files.

Reviewer #2: No

Reviewer #3: **Yes: **Tahrir N Aldelaimi

---

## [Editor Report · Acceptance letter]

11 Jul 2024

PONE-D-24-16749R1 

PLOS ONE

Dear Dr. Chi, 

I'm pleased to inform you that your manuscript has been deemed suitable for publication in PLOS ONE. Congratulations! Your manuscript is now being handed over to our production team.

Kind regards, 

on behalf of

Dr. Hadi Ghasemi 

Academic Editor

PLOS ONE